# The Impact of Service Learning on Academic, Professional and Physical Wellbeing Competences of EFL Teacher Education Students

**DOI:** 10.3390/ijerph20064852

**Published:** 2023-03-09

**Authors:** Eeva-Maria Hooli, Silvia Corral-Robles, José Luis Ortega-Martín, Antonio Baena-Extremera, Pedro Jesús Ruiz-Montero

**Affiliations:** 1Department of Didactics of Language and Literature, Faculty of Education, University of Granada, 18071 Granada, Spain; 2Department of Didactics of Musical, Plastic and Body Expression, Faculty of Education, University of Granada, 18071 Granada, Spain; 3Department of Physical Education and Sports, Faculty of Sport Sciences, University of Granada, 18011 Granada, Spain

**Keywords:** service learning, teaching English as a foreign language, physical wellbeing, competencies, socially disadvantaged groups, university education

## Abstract

In response to the challenges of the 21st century, the European Higher Education Area (EHEA) has been committed to the development of a training model that focuses on the acquisition of cognitive, physical, and social competences, among others, rather than the mere acquisition of knowledge. This approach has gained momentum in recent years, where the learners are the protagonists of their own learning process. This change of approach requires a change in methodology and involves a renewal of the methodological approach in Spanish universities. Service learning (S-L) is an active methodology that is gaining ground across universities due to its experiential, community-based and reflective characteristics. The present study aimed to provide an overview of the impact of S-L by active programs (physical activities, movement games, active tasks, etc.) on the acquisition of professional, linguistic, pedagogical or intercultural competencies, as well as physical wellbeing skills, on English as a foreign language (EFL) teacher education students. Fourteen Spanish EFL university students carried out an S-L active intervention with a migrant group from the Migrant Temporary Stay Centre in the autonomous city of Melilla (Spain). A qualitative study was designed to evaluate the achievement of these competencies. The results show that even though S-L is a challenging methodology, it favours the development of academic, professional, and physical wellbeing competences to succeed in a competitive and changing world, as well as the improvement of the participant students.

## 1. Introduction

In the field of higher education, new educational lines are beginning to emerge with the aim of responding forcefully to current social challenges. Some of these proposals include initiatives that link university education with socio-community development, physical and healthy attitudes, and social actions [1,2]. This fact opens the door to initiatives that can be carried out through active and experiential pedagogical models that are capable of contributing to better training of student teachers and an improvement in the quality of life of different social groups [2,3].

The current expansion of service learning (S-L) is noticeable in a multitude of disciplines and educational levels, with a high number of studies investigating its effects [4]. A great number of professionals and researchers in the education area, including representatives from the field of language teaching, especially English as a foreign language [5,6], contributed to this rapid growth. Therefore, S-L is presented as an effective methodology that promotes social values. According to Kirkgoz [7], university S-L and English as a foreign language (EFL) teaching help to consolidate knowledge through direct work with vulnerable and needy groups in real contexts.

Along this line, previous studies on university S-L [8,9,10,11,12] offered a panoramic view in terms of research and implementation. However, at the current point of the proliferation of this method, there is a need to provide specific analyses from each discipline and educational level [13,14].

With regard to the teaching of EFL and the S-L methodology for university students, it was shown that there is an increase in knowledge regarding the contents worked on, as well as the perceived academic and professional competencies [2,5]. In particular, it strengthens the idea that S-L represents a valuable opportunity to reinforce curricular content and various competencies covered in EFL teaching [2,15]. Other authors also analysed the effect of S-L, both at academic and professional levels, on the development of EFL teacher education students [15,16]. In contrast, other studies focused on the impact of this pedagogical method from a social point of view [17].

An S-L intervention with university students, who live in a multicultural context, may promote a very enriching intercultural experience through active programs with physical activity and games as the main tools. An S-L intervention with teacher education students specialising in EFL teaching involves taking into account various pedagogical, academic and professional elements [18]. One of these elements is university students’ own learning, which seems to be fundamentally enhanced by the value of the practical experience provided through S-L [19]. Similarly, small-group intervention in EFL teaching produces the consolidation of curricular learning in a comprehensive and globalised way, specifically with active methodologies that use active programs to be near students or any group [20]. Recent studies indicated that S-L facilitates the acquisition of knowledge related to a more inclusive pedagogy in EFL teaching with the strategies they implement during interventions, especially with regard to social, healthy educative and cultural values [2,18].

Other studies argued that S-L is a useful tool for EFL teaching to experience real-life situations: it awakens in them an interest in working with people from different backgrounds and characteristics (cultural, religious, ethnic, etc.) [21]. In terms of the educational, professional and physical wellbeing competencies acquired, there is a real action context in which the participating students play a decisive role in the development of these competencies in EFL teaching [20,22]. Another important variable is the critical reflection of university students in English language teaching programmes to establish a motivation that helps them to develop professional competencies more effectively [23].

Physical wellbeing is a healthy habit that must not be forgotten by university students of any area [24]. Is well known that during the higher education period, there is a multitude of important phases of development (physical, psychological, social, etc.) that could affect different academic career chapters of the students, such as competition between peers, the amount of academic work or changes in living conditions [24,25]. Moreover, Ghassab-Abdollahi et al. [26] emphasised the students’ low level of quality of life due to several factors, such as satisfaction with their studies [27], physical activity level [28], social relationships [29] and emotional wellbeing [30]. These factors and others have an effect on the effectiveness of teaching, especially on EFL teacher education students [2]. In addition, active experiences of EFL teacher education students with vulnerable groups through physical activity or movement games might produce an improvement in physical wellbeing because of the need for movement during the activities [24].

Therefore, in order to maximise the benefits of S-L at the professional learning level and physical wellbeing competence, the relationship between the academic content and the objectives of the intervention must be made explicit. Taking all of the above into account, the aim of this study was to show the impact of S-L via active programs (movement games, storytelling, physical activities, traditional games, gross motor activities theatre performances, etc.) on the acquisition of professional, linguistic, pedagogical, intercultural and physical wellbeing competencies on English as a foreign language (EFL) teacher education students.

## 2. Materials and Methods

### 2.1. Participants

A total of 14 EFL university students (11 female and 3 male), taking a Degree Program in Primary Education in the Faculty of Education and Sports Sciences in the Campus of Melilla of the University of Granada, participated in an S-L intervention with a minor migrant group in the Migrant Temporary Stay Centre in the autonomous city of Melilla (Spain). The ages ranged between 20 and 25 years old (M = 21.5, SD = 1.45). Moreover, lecturers who were keen to contribute to help in vulnerable contexts were selected to participate in the project and the vulnerable groups chosen were migrant children (4–12 years old) and teenagers between 13 and 17 years old. For this study, the subject chosen was “Teaching English Literature for Children and Young People” included in the last course of the aforementioned degree (2019/2020 academic year). This subject was based on teaching English literature through storytelling, traditional and movement games, theatre performances and gross motor activities by using books such as “The Adventures of Tom Sawyer”, “The Jungle Book”, and “Harry Potter”.

### 2.2. Design

The intervention programme was structured in two 45–50 min sessions per week and was carried out over four weeks (Figure 1). The teacher education students could spend up to 15 min preparing the materials and organising the group before the class; thus, the total amount of class time could reach 60 min. The teacher education students worked in pairs and the recipients were split into groups of four or five members.

As previously described, the subject within which the interventions took place was “Teaching English Literature for Children and Young People”. However, it is important to highlight that a physical and playful approach was employed to teach these academic contents. The interventions were designed by the university students in class together with the lecturer prior to carrying them out within the S-L programme. The university students had two lessons per week; one lesson focused on a theoretical approach and the other on a more practical one. The theoretical lesson was employed to prepare the materials for the S-L interventions carried out at the Migrant Temporary Stay Centre.

All the groups worked in the same space but separately, as each group used different approaches, materials and contents. Every session was designed and planned under the supervision of the teaching staff responsible for the university courses. The university students presented their intervention proposals during the weekly theoretical class before going to the Migrant Temporary Stay Centre, where the practical class took place. By doing this, the university students learned curricular content by designing, teaching and assessing EFL tasks provided to vulnerable groups.

The S-L programme was aimed at offering another perspective on teaching through the university student’s own experience. Moreover, the university students involved in the S-L intervention got to be familiarised with the real educational needs of the city while they provided educational services to migrants. At the same time, these students promoted social inclusion and education to vulnerable groups through physical activities and movement games. Another aspect to be considered was that the minor migrants and teenagers hardly knew any English, and many of them only spoke their mother tongue. Thus, the English level was low, and the migrant group and university students used Spanish or other languages, if they knew any, as a pivot language, using translanguaging as a pedagogical strategy [31].

The ethical criteria and good practices established by the University of Granada (Spain) for research were faithfully followed (code no. 1732/CEIH/2020). In the development of the S-L intervention, informed consent and permission were sought and freely given by the participants. They were duly informed about the objectives and topics to be addressed, preserving their identity above all else.

### 2.3. The Reflective Diary as a Data Collection Technique

A reflective diary was used as an instrument for data collection. The reflective technique was selected to evaluate the current S-L experience as it helps to improve personal, social, academic and physical wellbeing skills in the involved students [9,32]. Moreover, the reflective technique allows for revealing university students’ daily experiences [33]. This is especially significant in the practical classes through S-L intervention with socially disadvantaged groups as migrants [13]. These techniques allowed us to understand, through the experiences expressed in the participants’ diaries, the presence and/or absence of academic and professional competencies.

University students reflected on the problems they encountered and unexpected events or incidents that occurred during the development of the physical activities and movement games with the migrant unaccompanied minors, and on their ability to react to them. It was intended to provide university students with resources that would help them develop critical thinking skills [34]. This was done by posing four open questions [35]: (a) a description of an event that occurred during the S-L intervention through physical activity and movement games, (b) the reaction or response to the event, (c) determining the best response to the event, (d) the origin of the event and (e) the ability to respond to future events based on knowledge of the event in question. The university students completed their personal reflective diaries on a weekly basis and sent them to the lecturer through the course management system. The procedure used to transcribe the reflective diaries and to protect the participants’ anonymity was to assign a code to each student, always respecting the gender (female—f, male—m) and the number on the class list (from A1 to A14). If the first student in the class list was female, the code would be Af1. A total of 14 reflective diaries were written, and a total of 38 pages were obtained from the transcription of these reflective diaries. All reflective diaries were written in Spanish and then translated into English to be included in this study. It is necessary to highlight that two members of the research team participated as lecturers in the project. Thus, the absence of research bias was not possible; however, steps were taken in order to reduce the impact of bias in the findings. An example of this was that analysis of the reflective diaries was carried out with the supervision of a lecturer external to the course subject involved.

### 2.4. Data Analysis Process

This study was carried out through a qualitative content analysis, given that it was crucial to capture the essential aspects of the social phenomena presented from the viewpoints of the participants in the study [36]. Content analysis is an appropriate set of techniques used for analysing trends in communication content, such as the narratives of study participants. This analysis is carried out through a deductive–inductive approach. This means that there are two sources of information: the existing literature related to the topic of the study (for the deductive approach) and the narratives of the students (for the inductive approach).

Regarding the deductive approach, and therefore, the existing literature, García [31] and Kubanyiova [37] were two of the main authors studied that contributed to this approach of the analysis. García [31] presented the theory of language teacher education, identifying four main types of knowledge that a teacher should master to cope with a language classroom: (a) knowledge of the language, (b) knowledge about the language, (c) pedagogical knowledge and (d) intercultural understanding. More recently, Kubanyiova [37] highlighted the necessity of a teacher education student that addresses the new L2 teacher roles demanded by a diverse and multilingual society. Moreover, it is important to put emphasis on their physical wellbeing competence and the efficacy of physical activity and movement games as useful tools for learning EFL.

After implementing the deductive approach, the inductive approach was employed. In this inductive approach, the narratives of the student teachers were the main source of information. This meant that the researchers were responsible for analysing these narratives (data) to develop conclusions (explanations and interpretations). To do this, we needed to familiarise ourselves with the data. Familiarity with data includes reading the narratives several times to immerse oneself in the data. This can be described as a holistic content-reading process. During this process, we identified the main themes the student teachers were describing in the narratives. These themes corresponded to the different categories and subcategories.

This process of identification of categories and subcategories (themes) was done through the use of a cyclical process around three fundamental stages: discovery, codification and relativisation of the data [38]. The first stage allowed us to discover emerging topics in the text by breaking them into relatively small content units, then submitting them to descriptive treatment [39]. The second stage allowed us to carry out the categorisation and coding of these data in the N-Vivo qualitative analysis software. This part of the process was done in order to determine the trends and patterns, structures and discourses of communication.

Apart from determining the trends and patterns, through the use of this analysis software, it was possible to determine the frequency of appearance of the themes in the narratives of the students, that is, we could quantify the presence of the themes that emerged in their narratives [40]. Thus, it was possible to know the number of references or mentions the students made to the different themes (categories and subcategories) in their narratives. The number of references made to each category and subcategory are displayed in the tables presented below in the Results section. In this second stage, this data transformation of the qualitative analysis was reviewed by four experts, who paid attention to the credibility, transferability and confirmability of the data. This allowed us to make sure of the trustworthiness and validity of the study. In the third stage, the data were interpreted in order to formulate conclusions. In this step, all the information was prepared to capture the richness of the data and to convert different researchers’ interpretations into results.

As a result, a definitive system of categories and subcategories was created, as can be seen in Table 1.

## 3. Results

This section presents the results obtained from the analysis of the diaries of the university students under study. For a better understanding of the results, five action lines were drawn corresponding to the main categories and subcategories extracted. These categories and subcategories contributed to a better understanding of the competencies developed by university students during and after the implementation of the S-L experience with a group of unaccompanied minor migrants in Melilla [41]. The examples provide an overview of the experiences and perceptions of the university students; however, in order to respect their privacy and respect their rights, numerical codes were assigned as previously mentioned.

### 3.1. Pedagogical Competence

The dimension “Pedagogical Competence” was made up of three categories, as can be observed in Table 2. This dimension was based on sound, broad and current knowledge within the subject area, namely, English, as well as within the subject-based teaching and learning aspects, such as active methodologies, where physical activities and movement games played a key role; digital and material resources; and classroom management strategies implemented through a student-centred approach. In their narratives, the student teachers referred to these pedagogical aspects on 182 occasions, as can be seen in the following table.

The “Classroom Management” category presented the highest number of references, with 86 references in total. This means that the university students wrote about different aspects related to classroom management 86 times. They declared that they learned many different management strategies, for instance, how to manage silence and disruptions, time organisation, collaborative work and content adaptation while they were implementing the S-L intervention with unaccompanied migrant children.

The subcategory “silence and disruptions” presented 29 references from the total of 86 references. From their testimonies, it can be observed that they had some difficulties managing the classroom, as not all of them understood their new environment and may have had adaptation problems.

Am8: “The hardest part we encountered was to create an atmosphere of concentration in the classroom since they presented negative patterns of interaction”.

As they planned different physical activities and language games for their English classes, they thought it could be a good option to use some relaxation activities by implementing the Suggestopedia method.

Am8: “In the second session, I tried to do a relaxation exercise at the beginning of the class to relax them and create a better atmosphere”.

In the subcategory, “planning and organization”, a total of 25 references were found in the university student’s narratives. They realised how relevant it was to plan and organise their physical activities and language games sessions in detail and to always have more activities as a backup plan.

Af1: “In order to make the best use of time we need to plan our sessions better. It is also important to always have ‘back-up’ plans, as at any moment you may not be able to use what you had planned”.

Af4: “I could realise that having my sessions and activities planned in advanced, I could manage the time of the lesson in a better way and minimise certain discipline problems”.

The following category with the highest number of references was “Methodologies”, with a total of 65 references. This referred to the methodologies used by the university student teachers in their S-L interventions. In their testimonies, the play-based methodology presented a relevant role as university students became aware of the numerous opportunities offered by this methodology for teaching a foreign language [42]. Through the use of movement games and physical activity, a better, stress-free climate is created, which improves motivation [2,43], and it helps minor migrants with a low level of proficiency not to feel frustrated due to their lack of understanding of the language [44].

The “play based-methodology” subcategory obtained a total of 29 references. This can be seen in the following examples:

Af4: “I have learned that children of this age need more playful and entertaining activities in order to really learn the content”.

Af7: “After the first sessions, I realised that a playful mindset for children helps them engage in formal learning where they are motivated, they accept better the mistakes and are open to try out something new”.

The subcategory related to the “Total Physical Respond method” or TPR method presented 23 references from a total of 64 references. This is a teaching method that revolves around the coordination of speech and student action. Its foundation is teaching and learning through physical action, utilising the students’ motor skills while they respond to commands as quickly as possible. The university students reflected on the benefits of this active methodology, as illustrated in the following examples:

Af2: “The children responded much better to the dynamic activities where movement was used. By singing and dancing the students understood better the concepts, even if they were not familiar with the vocabulary at the beginning”.

Af1: “After implementing the TPR method by using the physical activities, I could see the students were developing the speaking and listening skills in record time”.

### 3.2. Linguistic Competence

The dimension “Linguistic Competence” was related to the ability of what a language user or learner is able to do with a language in order to perform written and oral discussions [45]. There are three main skills that can be highlighted when talking about the competence of a language user when they are receiving play-based and physical learning activities: receptive skills comprising listening and reading; productive skills, including writing and speaking; and interaction that can be produced online or in person and mediation [46]. In this study, this dimension referred to the ability to promote the language skills previously mentioned in the target learners. The mediation skill was not covered in this study as there was no evidence in their reflections.

As can be seen in Table 3, for the “Productive Skills” category and the “Receptive Skills” category, 25 and 13 references, respectively, were obtained from the analysis of the university students’ diaries. These were the number of times students mentioned the different aspects related to these skills. In particular, the subcategory “speaking”, with 21 references, along with the subcategory “listening”, 11 references, were the ones most discussed. The category “Interaction” obtained only three references.

This is evidence that the student teachers were aware of the importance of implementing a communicative approach. This can be observed in the following examples where real situations were encouraged:

Af10: “After the presentation of the activity, we handed out a card with a map of Melilla. The exercise consisted of starting from one point (the port of Melilla) and arriving to another site (the market), using directions in English (go straight on, turn right...), they followed the path by drawing it on the map”.

Af12: “The third day, they performed different role-plays. The first one was a conversation between a waiter and a customer. The second was a dialogue between a salesman and a customer buying a product”.

### 3.3. Psycholinguistic Competence

The “psycholinguistic competence” dimension refers to the ability of teacher education university students to recognise developmental differences in several active situations. Student teachers need to be able to address psychological, emotional, cognitive and social differences in lessons [47], where the contact with minor migrants is close because of the main characteristic of physical activity and movement games [13]. This definition is related to the two categories presented in Table 4: “Emotional competence and Multiple Intelligences”. Both categories presented a large number of references, 32 for the first one and 25 for the second category.

The category “Emotional Competence” referred to the ability to deal with emotions and it comprised two subcategories: “Being self-aware of their emotions”, with a total of 20 references, and the “promotion of emotional competence”, where 12 references were coded.

The subcategory “being self-aware of their emotions” referred to the way student teachers felt more conscious of their own emotions after having experienced the S-L intervention. This made them feel better prepared for future situations in the classroom. These are some examples of it:

Af2: “We have really enjoyed this experience and we have also worked much better compared to the first session, we feel more confident and less nervous”.

Am6: “The most relevant thing I have learned in this experience is the importance of building my self-confidence. I could realise that this attribute is key to successful teaching practices”.

The subcategory “promotion of emotional competence” referred to the implications of the university student to promote the general welfare of the minor migrants in the English classroom. In this process, the use of physical activities and movement games was key, as can be observed in the following examples:

Af14: “The children were very receptive and enthusiastic to participate in all the activities, but there were some who were perhaps shyer and more reserved and found it more difficult to be integrated with the group. To get them on board, we played an inclusion game that worked very well”.

Af: “Every time I felt that some of the students were having problems socialising with the group, I used movement games. These active dynamics and activities help them to feel relax while they were learning the target language”.

The category “Multiple Intelligences” presented a total of 25 references. Gardner’s theory introduced the idea of an individual with different intellectual capacities, that is, different ways of knowing, understanding and learning about our world. This theory was supported by numerous studies [48,49]. After having different experiences in the S-L intervention, the reflections and experience-sharing allowed the university students to be more conscious of the importance of recognising the multiple intelligences of the students. The following quotations are representative:

Am3: “I have learned that in a future I would broaden my ‘range’ of activities, making them more dynamic and, above all, bringing them closer to the children’ characteristics. Always bearing in mind that the main objective is that the student enjoys and learns the language”.

Af1: “In this experience I have learned to observe and notice children’s behaviour. Everyone has a different personality and behaves in a different way. Thus, as their teacher I need to help them to learn despite their different needs”.

### 3.4. Intercultural Competence

Culture is a key feature, which goes hand in hand with language teaching. The “intercultural competence” dimension refers to the “ability to ensure a shared understanding by people of different social identities, and their ability to interact with people as complex human beings with multiple identities and their own individuality” [50]. As can be seen in the following Table 5, two categories emerged: multiculturalism and translanguaging.

In this study, intercultural competence was key as the target group of the intervention was a minor migrant group whose country of origin was mainly Morocco. They were children whose native language was Arabic and, as they arrived in Spain, they are in contact with the Spanish language. Thus, monolingual-oriented or even bilingual education was not a possibility for them. They needed to be treated as multilingual minors [31,51]. Examples of student teacher testimonies related to multiculturalism were the following:

Af7: “In a city like Melilla where different cultures coexist and a large part of the migrant students are not fluent in Spanish, it is really important to promote multiculturalism and multilingualism”.

Am3: “Thanks to this experience I have been able to enjoy teaching for migrant students. On this occasion, I learnt about the importance of knowing different languages and cultures to make these students more comfortable in the language class”.

The category “Translanguaging”, which presented a total of 33 references, referred to the idea of using all the university students’ linguistic and cognitive resources to help minor migrants with the different languages that coexist in the classroom. This linguistic knowledge can be used as a vehicle in some of the language games and physical activities to help the students feel more relaxed when they do not understand the target language [52,53,54]. This situation can be appreciated in the following excerpts:

Am3: “Luckily, my partner was fluent in the languages spoken in class (French, English, Arabic, Tamazight and Spanish), which made communication much easier”.

Af5: “A solution to the main problem would have been the knowledge of key words in the different languages seen in the classroom to improve the explanation of the physical activities and the movement games”.

Af9: “We tried to explain ourselves through the use of gestures and also with the help of the teacher who was in the classroom by employing some Arabic words. This situation helped me understand the importance of knowing different languages and being flexible about the use of the target language”.

### 3.5. Physical Wellbeing Competence

In this study, this dimension referred to the ability to improve the physical wellbeing of the EFL student teachers and the target learners through the employment of physical activities and movement games sessions. The physical wellbeing competence dimension presented 11 references and it comprised two categories: “Physical wellbeing for teachers”, with a total of 6 references, and “Physical wellbeing for students”, with a total of 5 references. This can be seen in the following Table 6.

In the following example, it can be seen how the S-L interventions influenced the physical wellbeing of the EFL student teachers and the target learners by providing language input through games and physical activity. They declared that this teaching approach benefited their psychological and physical wellbeing [3,13].

Am8: “I found physical activities and games are a powerful vocabulary builder and actually benefit students by helping them shape their attentiveness and motivation and training the brain in how to learn”.

Af2: “I think all the games and activities that we have been doing over this period have helped teachers and students to remain physically active”.

Af7: “I was surprised when some of the students told me they felt better after the activities and games. They felt they belong to the group and therefore, more secure and comfortable”.

## 4. Discussion

The aim of the present study was to show the impact of S-L using active programmes (children’s games, physical activities, traditional games, theatre performances, etc.) on the acquisition of competencies, such as professional, linguistic, pedagogical and intercultural competencies, as well as physical wellbeing on English as a foreign language (EFL) teacher education students.

As stated above, the unaccompanied migrant children came from very different geographical areas and left their home countries for very different reasons. They may have come to Spain to seek new life opportunities [55]. These children may suffer from short-term adaptation and potential long-term assimilation into society; thus, a way to make them feel that they belong to society is through education.

This section provides a discussion of the findings presented in the previous section, which presented an overview of the experiences and perceptions of the student teachers under study regarding the development of many different competencies through the implementation of the S-L methodology with a group of minor migrants in Melilla. The information presented in the five dimensions analysed contributed to showing the necessary competencies (academic, professional and physical wellbeing) required by teacher education students with a specialisation in English when teaching EFL.

Regarding the development of the pedagogical competence of the student teachers, it can be said from the results that classroom management was one of the aspects that university students highlighted the most. They stated that this experience, namely, the S-L intervention with unaccompanied migrant children, helped them to improve many different management strategies, such as management of silence and disruptions, time organisation, collaborative work and content adaptation. Almost all of the university students referred to the importance of learning practical classroom strategies to manage meaningful communication and interactions, as well as knowing how to break negative patterns of interaction. Such situations highlighted the importance of lesson planning in keeping the class moving smoothly from task to task regardless of the learning environment. They stated that improving their way of planning the lesson minimised the need for discipline and allowed them to make the most of their time with students.

This was supported by Lopes and Oliveira [56], who stated that clear guidelines, rules and routines need to be addressed to ensure that new designs do not detrimentally impinge on learning. Furthermore, they declared that when learning extends outside the traditional physical classroom, as was the case in this study when implementing physical activities and movement games, teachers are responsible for working within the boundaries of acceptable and safe behaviour.

Moreover, the employment of active methodologies, such as S-L, together with the use of play-based learning and physical activities with the total physical method enables students to develop this sense of confidence and belonging. Numerous studies [57,58,59] showed the benefits these methods provide in the language learning process of target learners, for example, they retain what they learn in the long term [60].

In relation to linguistic competence, which refers to the ability to promote the language skills previously mentioned, of the target learners, the student teachers under study focused their efforts on promoting two of the basic skills: speaking and listening. This was due to the fact that the level of English of the minor migrants was low and a communicative approach was the most suitable methodology for them [61,62].

By employing real situations during the active time, they could connect with their previous knowledge and generate significant learning, despite the different native languages or cultures the minors had [63]. This approach could help the target learners to acquire the language in a more natural way, have space to experiment with new ways of communicating their thoughts and feelings, and use their imagination through different physical activities and games. It allowed them to connect with cultural traditions from their communities and expose them to new ones.

These findings are on the same track as the relevant studies presented by Windle and Miller [64], who stated that this communicative teaching approach is particularly helpful for children who are coping with difficult experiences and need a break from thinking about them.

Psycholinguistic competence refers to the ability to identify emotions and feelings in the teaching and learning process. The results showed that teaching English through the use of physical activities and language games made the teaching and learning processes less stressful and promoted an environment of mutual help and reciprocal exchange [20,65]. For instance, some of the university students highlighted the importance of being conscious of the different emotions of the minor migrants to be able to involve them in their own learning process.

On several occasions, the student teachers mentioned the benefits of implementing a play-based method through the use of language games and physical activities to evoke efficient emotional experiences, not only for the target learners but for themselves as teachers. These findings are in line with the result of the study conducted by Plass et al. [66] and Greipl et al. [67], who found that the implementation of games and physical activities facilitates learning by fostering teachers’ and learners’ cognitive, affective and emotional engagement.

After implementing this S-L intervention, the student teachers suggested an improvement in the learners’ and teachers’ capacity for emotional self-regulation. These results could imply that S-L may be a useful tool for improving EFL educators’ professional competencies.

In this study, intercultural competence was key, as the target group of the intervention was a minor migrant group. Thus, monolingual-oriented or even bilingual education was not a possibility with them. They needed to be treated as multilingual minors [31,51]. However, the shift towards plurilingualism is a challenge when having traditional conceptions and practices within language teaching and learning.

As Jhingran [68] stated, decades of research evidence indicates that most people in the world live in multilingual communities, and thus, it is important that schools encourage the use and further development of the languages and cultures students already know and the ones they need to know to meet the social demands in their host country. Multilingual programmes, pedagogies and systems are particularly important for students from migrant and refugee backgrounds.

In this study, the student teachers referred to this experience as a turning point for them, as they could be aware of the importance of applying a multilingual pedagogy and being flexible regarding the use of the target language, as well as promoting the native language of the students at the same time.

The studies of Franceschini [69] and French [70] presented similar results and mentioned the great opportunities multilingualism and translanguaging provide, not only for the target learners but also for teachers. They also concluded that active methodologies and student-centred approaches, such as the ones used in this research, are good mechanisms for cultural and social inclusion.

As was mentioned in the Results section, the physical wellbeing competence dimension referred to the ability to improve the physical wellbeing of the EFL student teachers and the target learners through the employment of physical activities and movement games sessions.

Regarding this dimension, it is important to highlight that most of the student teachers made a relevant reflection on the diaries stating that by participating in the S-L programme, they became more aware of the difficult situation in which the recipients of the service lived, generating a rethinking of their own wellbeing in comparison to those of others [71]. This reflection helped them to think more deeply about the need to make the minor migrants feel better physically and emotionally.

Thus, as the benefits of implementing physical activities and games in the language learning process are well known, they thought that this play-based approach was the most adequate one. After each session, good results could be seen, as shown in their testimonies. Moreover, the physical wellness feeling was not restricted only to the students since the student teachers could also feel and see a more effective way of teaching and all the benefits involved. These findings are on the same track as the research carried out by Curry and O’Brien [72], where they mentioned the importance of promoting physical wellbeing, not only for the target learners but also for the teachers.

## 5. Conclusions

To our knowledge, after reviewing the literature, no previous study carried out a similar analysis or addressed the same categories and subcategories as those in the present work.

However, the results obtained in this study must be carefully interpreted due to the various limitations. In the first place, the narratives were derived exclusively from teacher education students’ diaries, limiting their transferability. The use of the reflective diary by minor migrants involved in the S-L programme through physical activity and movement games might have provided a new perspective to the study. In addition, the study focused on S-L programmes that were conducted with a single vulnerable group. Consequently, we speculate that teacher education students might have a more significant impact on the community if they work with other vulnerable populations.

Despite these limitations, the main contribution of this study to the scientific knowledge in the field of higher education is the usefulness of S-L methodology on the different competencies required by teacher education students with a specialisation in English. It is also important to highlight that the EFL teacher education students were offered a unique opportunity to experience in a real context. At the same time, the migrant group from the Migrant Temporary Stay Centre taught them social competencies and the S-L programme content helped the students to achieve their academic, professional and physical wellbeing competencies.

This study also provides important information for academics and/or professors of EFL who are interested in applying S-L to socially excluded groups with group dynamics and games. Moreover, it contributed to a better understanding of the development of the academic and professional competencies of the teacher education students involved.

## Figures and Tables

**Figure 1 ijerph-20-04852-f001:**
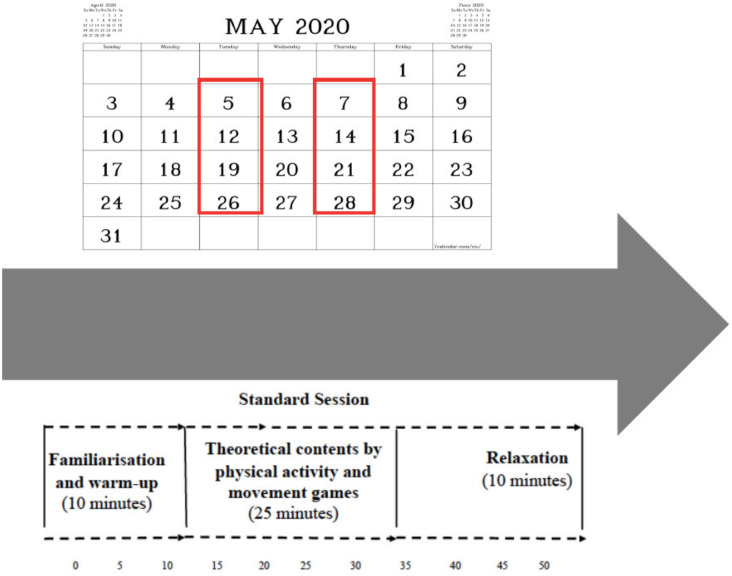
S-L intervention dates and an example of a standard session.

**Table 1 ijerph-20-04852-t001:** System of categories resulting from the teacher education students’ experiences.

Dimensions	Categories	Subcategories
Linguistic competence	1. Receptive skills	1.1 Listening
		1.2 Reading
	2. Productive skills	2.1 Writing
		2.2 Speaking
	3. Interaction	
Pedagogical competence	1. Classroom management	1.1 Silence and disruptions
		1.2 Time organisation
		1.3 Collaborative work
		1.4 Content adaptation
	2. Methodologies	2.1 Play-based methodology
		2.2 Total physical response
		2.3 Communicative approach
	3. Resources	3.1 Material resources
		3.2 Digital resources
Psycholinguistic competence	1. Emotional competence	1.1 Promotion of emotional competence
		1.2 Being self-aware of their emotions
	2. Multiple intelligences	
Intercultural competence	1. Multiculturalism	
	2. Translanguaging	
Physical wellbeing competence	1. Physical wellbeing of student teachers	
	2. Physical wellness of target learners	

**Table 2 ijerph-20-04852-t002:** Number of references for the pedagogical competence.

Dimensions	Categories	Subcategories	No. References
Pedagogical competence (182 references in total)	1. Classroom management (86 references—47.2% of the total)	1.1 Silence and disruptions	29
		1.2 Time organisation	25
		1.3 Collaborative work	12
		1.4 Content adaptation	10
	2. Methodologies (64 references—35.2% of the total)	2.1 Play-based methodology	29
		2.2 Total physical response	23
		2.3 Communicative approach	12
	3. Resources (32 references—17.6% of the total)	3.1 Material resources	26
		3.2 Digital resources	6

**Table 3 ijerph-20-04852-t003:** Number of references of linguistic competence.

Dimensions	Categories	Subcategories	No. References
Linguistic competence (41 references in total)	1. Receptive skills (13 references—31.7% of the total)	1.1 Listening	11
		1.2 Reading	2
	2. Productive skills (25 references—60.9% of the total)	2.1 Writing	4
		2.2 Speaking	21
	3. Interaction (3 references—7.4% of the total)		3

**Table 4 ijerph-20-04852-t004:** Number of references for psycholinguistic competence.

Dimensions	Categories	Subcategories	No. References
Psycholinguistic competence (57 references in total)	1. Emotional competence (32 references and 56.2% of the total)	1.1 Promotion of emotional competence	12
		1.2 Being self-aware of their emotions	20
	2. Multiple intelligences (25 references and 43.8% of the total)		25

**Table 5 ijerph-20-04852-t005:** Number of references for intercultural competence.

Dimensions	Categories	No. References
Intercultural competence (64 references in total)	1. Multiculturalism (31 references—48.4% of the total)	31
	2. Translanguaging (33 references—51.6 of the total)	33

**Table 6 ijerph-20-04852-t006:** Number of references for physical wellbeing competence.

Dimensions	Categories	No. References
Physical wellbeing competence (11 references in total)	1. Physical wellbeing for students (6 references and 54.5% of the total)	6
	2. Physical wellbeing for students (5 references and 45.5% of the total)	5

## Data Availability

The datasets generated during and analysed during the current study are available from P.J.R.-M. upon reasonable request.

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
