# Peer review of "The Impact of Service Learning on Academic, Professional and Physical Wellbeing Competences of EFL Teacher Education Students"

_ijerph, 2023, doi:10.3390/ijerph20064852_

Round 1

Reviewer 1 Report

In this study, the reflective diaries of 14 EFL students, who follow a bachelor's program in Primary Education, are analyzed in order to highlight the importance of the use of Service-Learning (S-L) methodology in higher education. The authors should make the following changes to the article:

 (1) Quotations from participants (A1-A14) used as examples for various categories should be more numerous, for example 2-3 for each category. A simple analysis of frequencies (according to their magnitude) has limited relevance. Further interpretations and a deeper insight into the reflections of the participants are needed. This study can be used as an example:

 Davies, M. L. (1997). Shattered assumptions: Time and the experience of long-term HIV positivity. Social Science & Medicine, 44(5), 561–571. doi:10.1016/s0277-9536(96)00177-3.

 (2) Participants were assigned codes from A1 to A14. It is not enough. Information is missing on variables such as age and gender (data can be presented in tabular form). There is no section entitled Participants, where certain information relevant to the study are presented (demographic and social characteristics of the participants).

 (3) References in tables should be doubled with percentages. See, again, the study cited earlier.

 (4) The names of the authors must be verified. For example, instead of Sparkes, A. C. is written Sparker, A. Not only is the name misspelled, but this author has two first names (Andrew C.).

(38. Sparker, A. Narrative Analysis: Exploring the Whats and Hows of Personal Stories. In Qualitative Research in Health Care (1st Edn), 600 edited by I. Holloway, 2005. Open University Press.)

 (5) In the tables, the "explanation" or specification „Source: Own elaboration” must be omitted. It's an obvious thing, so it doesn't need to be specified.

 (6) The section Results must be separate from the section Discussion. As it looks now, the interpretation of the results is superficial. It must be extended, deepened, nuanced and separated from the presentation of the results.

 (7) It must be clearly stated what is the contribution of this study to the scientific knowledge in the field.

Author Response

Comments and Suggestions for Authors

In this study, the reflective diaries of 14 EFL students, who follow a bachelor's program in Primary Education, are analyzed in order to highlight the importance of the use of Service-Learning (S-L) methodology in higher education. The authors should make the following changes to the article:

 (1) Quotations from participants (A1-A14) used as examples for various categories should be more numerous, for example 2-3 for each category. A simple analysis of frequencies (according to their magnitude) has limited relevance. Further interpretations and a deeper insight into the reflections of the participants are needed. This study can be used as an example:

Davies, M. L. (1997). Shattered assumptions: Time and the experience of long-term HIV positivity. Social Science & Medicine, 44(5), 561–571. doi:10.1016/s0277-9536(96)00177-3.

Response: Thank you for your suggestions and the example provided. It was of great use. We have added more quotations from the participants to each category. We restructured the sections of the results and discussion. These two sections have been split and we have gone through deeper interpretations in the part of the discussion.

 (2) Participants were assigned codes from A1 to A14. It is not enough. Information is missing on variables such as age and gender (data can be presented in tabular form). There is no section entitled Participants, where certain information relevant to the study are presented (demographic and social characteristics of the participants).

Response: According to recommendation of reviewer 1, we have added extra information about participants and described the process to assign the codes to university students’ participants.

 (3) References in tables should be doubled with percentages. See, again, the study cited earlier.

Response: We have clarified the references and percentages in all tables.

 (4) The names of the authors must be verified. For example, instead of Sparkes, A. C. is written Sparker, A. Not only is the name misspelled, but this author has two first names (Andrew C.).

(38. Sparker, A. Narrative Analysis: Exploring the Whats and Hows of Personal Stories. In Qualitative Research in Health Care (1st Edn), 600 edited by I. Holloway, 2005. Open University Press.)

Response: Thank you for your suggestion. We have reviewed all references and added two first names.

 (5) In the tables, the "explanation" or specification „Source: Own elaboration” must be omitted. It's an obvious thing, so it doesn't need to be specified.

Response: We have deleted all the specifications “Own elaboration”.

 (6) The section Results must be separate from the section Discussion. As it looks now, the interpretation of the results is superficial. It must be extended, deepened, nuanced and separated from the presentation of the results.

Response: Thank you for your suggestion. We restructured the part of the results and discussion. These two sections have been split and we have gone through deeper interpretations in the part of the discussion. We think that these two parts are clearer now.

 (7) It must be clearly stated what is the contribution of this study to the scientific knowledge in the field.

Response: We have added the main contribution of this study in the conclusion section.

Reviewer 2 Report

The Article “The Impact of Service-Learning on Academic, Professional and Physical-Wellbeing Competence of EFL Teacher Education Students by Physical Movement” interesting but, in my opinion, not very well written scientific article. The major strength of the study is this study practical application, and it is relevant for real situation. Nevertheless, it requires several changes before it will be published. There are some remarks concerning this article:

1.      My biggest concern is related to the research presentation. Research results and discussion presented in one part. It should be separated.

2.              Additionally, I would like to note that introduction part is a little bit too long and even some parts suits to the textbook, but not to the research article (e.g. lines 48-63).

3.            Study aims presented in different manuscript parts differently (in abstract and in lines 123-126).

4.           Research Results and Material and Methodology parts have been mixed (lines 174-176).

5.            I would suggest the Intervention design  to present in a scheme for better understanding study design. Additionally, it is necessary to mention in which language diaries has been presented. Moreover, I would suggest to include data about students age, study year, gender and native language.

6.            Study intervention place mentioned in two places in manuscript and they’re not the same, e.g. lines 85-86 introduction part (I would argue that is not necessary to present there this info) and lines 130-131.

7.           There are no links in the text to the tables in article. Additionally, there is no need to add at each result part table that is “Source: Own elaboration”.

8.              In text and in tables presented data with number of references. It is not clear what they mean. Additionally, in the text appears some coded data (like A-1) It is not clear what it is? Is it categories, subcategories or separate participants’ quotations? 

9.            Reference list for such kind of scientific article is appropriate, cited 66 references and 44 of them published in recent 10 years. However, some of references presented not correctly, not in accordance to all requirements, without DOI and etc. (e.g., 17, 27, 49, 50 etc.). Some references have not been presented in results and discussion part (e.g. 54).

Before publication, in my opinion, article must be improved.

Author Response

Comments and Suggestions for Authors

The Article “The Impact of Service-Learning on Academic, Professional and Physical-Wellbeing Competence of EFL Teacher Education Students by Physical Movement” interesting but, in my opinion, not very well written scientific article. The major strength of the study is this study practical application, and it is relevant for real situation. Nevertheless, it requires several changes before it will be published. There are some remarks concerning this article:

Response: Thank you so much for your comments and we have followed all your suggestions.

1. My biggest concern is related to the research presentation. Research results and discussion presented in one part. It should be separated.

Response: Thank you for your suggestion. We restructured the section of the results and discussion. These two sections have been split and we have gone through deeper interpretations in the part of the discussion. We think that these two parts are clearer now.

2. Additionally, I would like to note that introduction part is a little bit too long and even some parts suits to the textbook, but not to the research article (e.g. lines 48-63).

Response: Thank you for your recommendation. We have modified the Introduction and now is shorter than before.

3. Study aims presented in different manuscript parts differently (in abstract and in lines 123-126).

Response: Thank you very much for your suggestion. We have presented similar aim in the Abstract, end of the Introduction and beginning of the Discussion.

4. Research Results and Material and Methodology parts have been mixed (lines 174-176).

Response: Thank you very much for this suggestion, it was a good remark. We have reviewed and modified the section 2 and 3 of the manuscript.

5. I would suggest the Intervention design to present in a scheme for better understanding study design. Additionally, it is necessary to mention in which language diaries has been presented. Moreover, I would suggest to include data about students age, study year, gender and native language.

Response: Thank you very much suggestion. We have followed your recommendation and, scheme and extra information have been added to text.

6. Study intervention place mentioned in two places in manuscript and they’re not the same, e.g. lines 85-86 introduction part (I would argue that is not necessary to present there this info) and lines 130-131.

Response: We have deleted the first mentioned place as reviewer 2 recommended. Thanks.

7. There are no links in the text to the tables in article. Additionally, there is no need to add at each result part table that is “Source: Own elaboration”.

Response: Thank you for your suggestion. We have included different links to the tables in the text. Also, these specifications related to ‘source: own elaboration’ have been deleted.

8. In text and in tables presented data with number of references. It is not clear what they mean. Additionally, in the text appears some coded data (like A-1) It is not clear what it is? Is it categories, subcategories or separate participants’ quotations?

Response: We have clarified the references and percentages in all tables. In addition, we have explained the coded data.

9. Reference list for such kind of scientific article is appropriate, cited 66 references and 44 of them published in recent 10 years. However, some of references presented not correctly, not in accordance to all requirements, without DOI and etc. (e.g., 17, 27, 49, 50 etc.). Some references have not been presented in results and discussion part (e.g., 54).

Before publication, in my opinion, article must be improved.

Response: Thank you very much for all your comments. We have followed the template and rules from the Journal and the number of the publication or DOI was not of mandatory use. However, some of them are specified.

Reviewer 3 Report

The present study explored the effects of service-learning approach on EFL teacher education students’ professional, linguistic, pedagogical, intercultural and physical- wellbeing competence. The findings suggested that S-L may be a useful tool for improving EFL teachers’ academic, professional and physical-wellbeing competences for better cultivating the students. Overall, the paper is well-written article. The findings of this study suggested that S-L may be a useful tool for improving EFL educators’ professional competences. However, I would suggest the authors to reorganize the structure of the paper. My major concern and recommendations are listed below.

1. It seems that the authors misplaced the sections in the Results section, the 2.2 and 2.3 should be placed in the Methods section.

2. I would suggest the authors to move those tables from discussion section to the results section. In results section, the authors should display those tables and interpret the results.

3. In the discussion section, please add more discussion corresponding to each finding, such as discussion on previous findings or implications of future study.

4. Overall, I would suggest the authors to reorganize the flow of the paper, e.g., Methods, Results, Discussion.

Author Response

Comments and Suggestions for Authors

The present study explored the effects of service-learning approach on EFL teacher education students’ professional, linguistic, pedagogical, intercultural and physical- wellbeing competence. The findings suggested that S-L may be a useful tool for improving EFL teachers’ academic, professional and physical-wellbeing competences for better cultivating the students. Overall, the paper is well-written article. The findings of this study suggested that S-L may be a useful tool for improving EFL educators’ professional competences. However, I would suggest the authors to reorganize the structure of the paper. My major concern and recommendations are listed below.

1. It seems that the authors misplaced the sections in the Results section, the 2.2 and 2.3 should be placed in the Methods section.

Response: Thank you very much and we have followed your recommendation.

2. I would suggest the authors to move those tables from discussion section to the results section. In results section, the authors should display those tables and interpret the results.

Response: We have moved the tables and separated the Results and Discussion sections. We think that these two parts are clearer now. Thanks.

3. In the discussion section, please add more discussion corresponding to each finding, such as discussion on previous findings or implications of future study.

Response: Thank you for your suggestion. We restructured the sections of the results and discussion. These two sections have been split and we have gone through deeper interpretations in the part of the discussion.

4. Overall, I would suggest the authors to reorganize the flow of the paper, e.g., Methods, Results, Discussion.

Response: Thanks. We have reorganized the whole document.

Reviewer 4 Report

The present article addresses the issue of Service-Learning methodology as a tool for improving competences of English as a Foreign Language teacher education students. In particular, the most interesting perspective lies in the fact that such an experiment was conducted with a group of minor migrants.

I found the introduction well-balanced in terms of exhaustivity of context and lenght, able to clarify to those readers not aware about S-L interventions both what they are and why they are useful for.

Nonetheless, it suffers for a very poor English grammar editing, many typos are there, and this critique has should be considered for the entire article, as well.

Perhaps, the description of the methods is a bit lazy and superficial. There is no need to create a subsection if there are not at least a couple of them. Probably, I would suggest to split this section in two or three parts (as you consider clearer) which should address separately and more extensivley the following three components of the study: i) EFL teacher education students; ii) Migrant Temporary Stay Centre students; iii) S-L programme content. Moreover, since the exposition of the methodology attains to section 2, I expect in this section to find also an exhaustive explanation of the reflective diaries data collection technique, followed by necessary hints about how them were transformed into analyzable qualitative data.

Indeed, at the end you come up with section 3 called "Results" and section 4 called "Results and Discussion", maybe exactly beacuse your misleading ripartition of information troughout the sections.

Finally, if on the one side I found interesting the description of your results, on the other side I am afraid that conclusions would need a clearer description of what you learned comprehensively from the analysis itself. In other words, could you extensively address to what extent we can learn from this study the potentially significant impact of S-L? And also what are the most important information obtained for academics and/or professors of EFL?

Author Response

Comments and Suggestions for Authors

The present article addresses the issue of Service-Learning methodology as a tool for improving competences of English as a Foreign Language teacher education students. In particular, the most interesting perspective lies in the fact that such an experiment was conducted with a group of minor migrants.

I found the introduction well-balanced in terms of exhaustivity of context and length, able to clarify to those readers not aware about S-L interventions both what they are and why they are useful for.

Nonetheless, it suffers for a very poor English grammar editing, many typos are there, and this critique has should be considered for the entire article, as well.

Response: Thank you very much for your words. The paper has been proofread by an English native speaker and we think is clearer now.

Perhaps, the description of the methods is a bit lazy and superficial. There is no need to create a subsection if there are not at least a couple of them. Probably, I would suggest splitting this section in two or three parts (as you consider clearer) which should address separately and more extensively the following three components of the study: i) EFL teacher education students; ii) Migrant Temporary Stay Centre students; iii) S-L programme content. Moreover, since the exposition of the methodology attains to section 2, I expect in this section to find also an exhaustive explanation of the reflective diaries data collection technique, followed by necessary hints about how them were transformed into analyzable qualitative data.

Indeed, at the end you come up with section 3 called "Results" and section 4 called "Results and Discussion", maybe exactly beacuse your misleading ripartition of information troughout the sections.

Response: Thank you for your suggestion. We restructured the sections of the results and discussion. These two sections have been split and we have gone through deeper interpretations in the part of the discussion.

Finally, if on the one side I found interesting the description of your results, on the other side I am afraid that conclusions would need a clearer description of what you learned comprehensively from the analysis itself. In other words, could you extensively address to what extent we can learn from this study the potentially significant impact of S-L? And also, what are the most important information obtained for academics and/or professors of EFL?

Response: We have added the importance of these three components of the study in the Conclusion, improving this section and explaining better the impact of S-L on the students involved.

Round 2

Reviewer 1 Report

I agree with all the clarifications introduced in the text of the article.

Author Response

Dear reviewer,

Thank you very much for your help and comments.

Regards,

Reviewer 2 Report

Authors of this publication “The Impact of Service-Learning on Academic, Professional and Physical-Wellbeing Competence of EFL Teacher Education Students by Physical Movement” has taken into account most of my remarks made during my firsts review process.

1.     Service-Learning (S-L) abbreviation used only in abstract and not presented in introduction once again.

2.     In Figure 1 text could be presented in bigger letters.

3.     Links to the tables in the article should appear before the table, but not below (e.g. table 2).

4.     Still the point related to the data presented in the tables with number of references is not clear.  What kind of references it is here presented? (e.g. table 2).

Author Response

Dear Editor,

Please, find enclosed the revised version of our manuscript entitled, “The impact of Service-Learning on academic, professional and physical-wellbeing competence of EFL teacher education students” by Dr. Hooli et al., to be considered for publication in International Journal of Environmental Research and Public Health. We would like to gratefully thank the Reviewers for their thoughtful and constructive comments, which have undoubtedly improved the quality of our manuscript. We have carefully considered all of the suggestions, and have integrated them into the revised manuscript. Changes to the original manuscript have been incorporated by using green background. We believe our manuscript is now stronger as a result of these modifications. An itemized point-by-point response to the Reviewers’ comments is presented below.

1. Service-Learning (S-L) abbreviation used only in abstract and not presented in introduction once again.

Response: Thank you very much for your good comment. We have written "Service-Learning" the first time that appears in the Introduction.

2. In Figure 1 text could be presented in bigger letters.

Response: Thanks for the suggestion. We have written bigger the words of the Figure 1.

3. Links to the tables in the article should appear before the table, but not below (e.g. table 2).

Response: Thanks. We have improved the link to the table 2 and appears now just before to the Table.

4. Still the point related to the data presented in the tables with number of references is not clear. What kind of references it is here presented? (e.g. table 2).

Response: We have tried to explain this fact the best possible. So, we hope now that it will be more understandable. Thanks a lot.